# Rearing Pigs with Intact Tails—Experiences and Practical Solutions in Sweden

**DOI:** 10.3390/ani9100812

**Published:** 2019-10-15

**Authors:** Torun Wallgren, Nils Lundeheim, Anna Wallenbeck, Rebecka Westin, Stefan Gunnarsson

**Affiliations:** 1Department of Animal Environment and Health, Swedish University of Agricultural Sciences (SLU), P.O.B. 234, S-532 23 Skara, Sweden; torun.wallgren@slu.se (T.W.); anna.wallenbeck@slu.se (A.W.); rebecka.westin@slu.se (R.W.); 2Department of Animal Breeding and Genetics, Swedish University of Agricultural Sciences (SLU), P.O.B 7023, S-750 07 Uppsala, Sweden; nils.lundeheim@slu.se; 3Farm and Animal Health (Gård&Djuhälsan), Uddetorp Röda huset, S-532 96 Skara, Sweden

**Keywords:** welfare, swine, tail-docking, tail-biting

## Abstract

**Simple Summary:**

Tail biting is a common problem within modern pig production and is mainly an indicator of poor housing environment where the behavioural needs of pigs are not met. Tail biting causes pain and can result in infection, leading to reduced pig growth and reduced farm profits. In order to prevent tail biting, pigs are often tail docked, without pain relief, within the first week of life. The EU Directive condemns routine tail docking and advises that tail biting can be prevented through improving the environment of pigs. In Sweden, tail docking is banned and all pigs are reared with intact tails. This paper summarises knowledge from Swedish production of undocked pigs and describes practical solutions in use in Sweden that can be applied to pig production in other EU Member States. Housing conditions and management within Swedish pig production, such as stocking density and feeding space, differ in many aspects from those in other EU countries. To prevent tail biting and eliminate the need for tail docking, EU legislation should more clearly match with the biological needs of pigs.

**Abstract:**

Tail biting is a common issue within commercial pig production. It is mainly an indicator of inadequate housing environment and results in reduced health welfare and production. To reduce the impact of tail biting, pigs are commonly tail docked, without pain relief, within the first week of life. EU Council Directive 2008/120/EC prohibits routine tail docking, but the practice is still widely used in many Member States. Sweden has banned tail docking since 1988 and all pigs have intact tails, yet tail biting is a minor problem. This paper summarises and synthesises experimental findings and practical expertise in production of undocked pigs in Sweden and describes solutions to facilitate a transition to producing pigs with intact tails within intensive pig production in the EU. Swedish pig housing conditions and management differ in many aspects from those in other EU Member States. Swedish experiences show that lower stocking density, provision of sufficient feeding space, no fully slatted flooring, strict maximum levels for noxious gases and regular provision of litter material are crucial for success when rearing pigs with intact tails. To prevent tail biting and to eliminate the need for tail docking, we strongly recommend that EU legislation should more clearly match the biological needs of pigs, as is done in Swedish legislation.

## 1. Introduction

Tail biting is a common issue within intensive pig production [1,2]. It is well documented that tail biting is an indication of an insufficient housing environment, leading pigs to redirected exploratory behaviour and subsequently manipulation and biting of tails [1,3,4,5,6]. Tail biting impairs pig health, production and welfare [7,8,9]. To reduce the impact of tail biting, pigs worldwide are commonly tail docked, without pain relief, within the first week of life [10]. The impact of rearing environment and use of, e.g., house enrichment to reduce tail biting has been thoroughly investigated for decades [3], but only a few countries with minor pig production in comparison to other EU countries, such as Sweden, actually rear undocked pigs [3,11,12].

Instead of providing a better production environment, 90–95% of the pigs produced within the European Union (EU) are tail docked. This is despite Council Directive 2008/120/EC [13] prohibiting routine tail docking and evidence that tail docking merely removes the symptoms of tail biting, not the underlying cause [14]. The Council Directive requires that pig housing environments should be improved to reduce tail biting has been largely ignored [11,15,16]. Furthermore, the fact that tail docking is commonly performed without any pain relief results in acute and sometimes long-term pain [17,18]. In countries that rear pigs with intact tails (e.g., Sweden and Finland), national legislation or other specific regulations or production schemes are normally the driving force, rather than the Council Directive [3].

The Swedish Animal Welfare Act of 1944 permitted tail docking, although in practice the procedure seemed to be rarely performed [19,20,21]. In 1988 all surgical interventions that cannot be justified from a veterinary point of view (including tail docking) were banned [22]. This law has been carried over into the latest Animal Welfare Act of 2018 [23].

The ban on tail docking is accompanied by other differences in the regulations on pig husbandry in Sweden compared with other EU Member States, regarding, e.g., space allowance and requirements for enrichment material (Table 1). Furthermore, Sweden has banned fully slatted floors in pig housing, whereas in other EU Member States pigs are commonly housed in larger groups in fully slatted pens [3,24]. Swedish legislation demands a more generous minimum space allowance, to enable all pigs to lie down at the same time (details in Figure 1). The legislation also covers factors such as eating space per pig depending on the live weight (LW), to ensure that it is physically possible for all pigs to eat at the same time in order to avoid competition at feeding, additional requirements on ventilation (e.g., maximum levels for ammonia and carbon dioxide gases) compared with the EU-legislation [25].

This paper summarises and synthesises results from experiments and practical experiences from Swedish production of undocked pigs. The aim is to identify practical solutions that can facilitate transition to producing pigs with intact tails within intensive pig production in the EU.

## 2. Descriptive of Results of Tail Biting in Sweden

Tail damage recorded at slaughter is an indicator of tail biting, although tail damage can also originate from issues other than tail biting, such as tail necrosis due to toxins in straw. It is commonly impossible to distinguish between the different origins of tail damage in the abattoir, so tail biting can be over-estimated, although abattoir data can underestimate the true prevalence of tail biting due to a blunt scoring system [26]. Moreover, scoring schemes differ between countries and assessors, making it difficult to compare data on tail biting from different sources [26,27]. To be scored as “tail bitten” in a Swedish abattoir, at least half the tail should be missing or show evident tail damage, or the carcass should be discarded due to abscesses caused by tail lesions [28]. Tail biting behaviour can give rise to less severe damage, such as swelling or bite marks, which are not likely to be detected at the abattoir but still affect welfare and production. In Sweden and Norway, where tail docking is banned and routine records of tail damage are collected at slaughter, tail biting is reported to vary between 1 and 3% [29,30] and lie around 4% [31], respectively. The definition of tail damage may differ between countries and over time. In 2018, 3.2% of the 2,471,524 pigs slaughtered in Sweden were scored as tail-bitten at the abattoir (Farm and Animal Health, unpublished data from routine recordings from herds associated with the advisement company Farm and Animal Health). Estimates of tail damage prevalence in pigs with undocked tails in other countries are based on short-term studies. A survey of six slaughterhouses in the UK involving 62,971 pigs [32] found that 9% of undocked pigs had damaged tails and that 0.5% of these pigs had the most severe form of damage (i.e., parts of the tail missing and severe wounds with swelling and signs of infections) [33]. A study at a single slaughterhouse in Finland found the total prevalence of tail damage in 10,852 pigs recorded was 34.6% and that the prevalence of fresh tail lesions and severe damage was 11.7% and 1.3%, respectively [7]. In a Swedish study that involved detailed observations of tail damage and tail length on 15,068 pigs slaughtered at two slaughterhouses, the prevalence of injury or shortening of the tail was found to be 7.0% and 7.2% in the two slaughterhouses, respectively [26]. When only considering pigs with half or less of the tail left, the values were 1.5% and 1.9%, respectively, which is in line with the prevalence estimated from the routine recordings at slaughter in Sweden [26]. The definition used in routine classification of tail damage in Sweden is “at least half of the tail missing or clear signs of bite damage on the tail” [28]. Thus, routine recordings at slaughter are an indication of tail damage, but the true prevalence, including both severe and milder forms of tail damage, is higher. Organically raised pigs in Sweden have been found to have approximately half the prevalence of damaged tails at slaughter compared to conventionally produced pigs [33], although data from organic pig production seem to fluctuate in recent years [34].

Swedish farmers’ opinions about tail biting and suspected causes were recently gathered in a telephone survey involving interviews with 60 farmers [35]. Tail biting was reported to occur (at least one outbreak) on 50% of growing pig farms (10–30 kg LW) and on 88% of finishing pig farms (30 kg LW-slaughter). On affected farms, tail-bitten pigs were commonly seen less than twice a year among growers (78.3% of farms) and 3–6 times a year among finishing pigs (37% of farms). In general, one pen in each affected batch had tail-bitten pigs. Finishing pig farms reported on average 1.6% (0.1–6.5%) tail-bitten pigs scored at the abattoir [35].

The most commonly suspected cause of tail biting reported by grower farmers was salt deficiency and too-high stocking density among grower farmers and composition/feed equipment malfunction and unknown causes among finishing pig farmers [36]. Identifying and removing the biter and separation of the bitten pig were the most common responses to tail biting outbreaks, followed by increasing the straw ration, checking ventilation and provision of other manipulable materials or toys [35]. However, 5% of farmers surveyed took no action at tail biting outbreaks [35]. It should also be noted that emergency docking is not practiced in Sweden.

## 3. Internal and External Factors in Relation to Tail Biting (and Contrasting Production Conditions)

### 3.1. Internal Factors

Gender has an impact on tail biting behaviour in pig groups. Swedish studies on pig groups with mixed genders have found that barrows more often are victims of tail biting [26,36] and that females are biters in more cases [8,36].

In other countries, tail biting has been demonstrated to increase with age [4,37]. This has also been found in Sweden [35,38,39] and the increase probably relates to the decreased space availability for pigs and to gilts coming into (pre)oestrous (approx. 4–5 months) and rejecting males.

Due to the sporadic nature of tail biting, there are difficulties in investigating the genetic background of the trait. Results in previous studies contradict each other, with some indicating that breed differences exist and other that they do not. This discrepancy is possibly due to small sample size, differences in animal material (breeds) between studies and differences in the definition of tail biting. A Swedish study analysing over 3000 Swedish Landrace, Yorkshire and Hampshire boars from a boar testing station showed that Landraces were more often tail biters, Yorkshires were more often receivers and Hampshires received less tail biting than both other breeds [40].

A study on the heritability of performance of tail biting found it to be 0.27 in a Landrace population [41]. The study also found indications of unfavourable genetic correlations between tail biting and lean tissue growth rate and back fat thickness. However, in the same study, tail biting was not found to be heritable for Large White pigs [41]. Taken together, the evidence in the scientific literature indicates that tail biting is to some extent influenced by pig genetics and that it is indirectly negatively affected by breeding for high lean meat growth.

### 3.2. External Factors

#### 3.2.1. Stocking Density

European Union legislation determines stocking density depending on weight categories and is presented in Figure 1 [13,42]. Swedish legislation regarding stocking density for growing pigs is instead determined by the actual LW of the pigs in the pen by the formula 0.17 + (live weight (kg)/130) [25]. In reality, this means that pigs reared under minimum requirements in other EU countries have less space than pigs reared under minimum requirements in Sweden until they reach slaughter at ~110 kg LW (Figure 1). The difference in stocking density between Swedish and EU legislation is largest at around 80–110 kg LW, which is also the period when most tail biting occurs [13,25]. According to the Swedish formula, 0.78 m^2^/pig needs to be provided for pigs at 80 kg LW and 0.94 m^2^/pig at 100 kg LW in Sweden. To fulfil this requirement, pigs are kept in pens of 0.8–0.9 m^2^/pig from the beginning of the finishing period (at approx. 30 kg LW). Pigs from the same batch are sent to slaughter on different occasions. After approximately 60–70 days, the largest 1–2 pigs in each pen are sent to slaughter. The same procedure is repeated two to three times until at last all remaining pigs are sent to slaughter. So if 13 pigs are put in the pen in the beginning, only 10 or less are left at the end of the rearing period. According to EU Council Directive 2008/120/EC, other parameters such as stocking density should be revised before tail docking is carried out [13]. The extent to which stocking density is manipulated to prevent tail biting in commercial production is unknown [13]. With an increasing number of piglets born and weaned from today’s highly prolific sows, all available space is typically already in use. Hence, reducing stocking density is difficult unless farmers are willing to reduce the actual number of sows in production (fewer pigs per unit area) or build new facilities (more space per pig).

#### 3.2.2. Group Size

A wide range of housing systems are used in the EU, but pen group size is generally large (24 pigs/pen or more) [3]. In Sweden, pens for both growers and finishers typically hold 10–13 pigs. The small group size has several benefits that could be of relevance in successful rearing of pigs with intact tails. One is that the lower number of pen mates limits the number of potential victims at a tail biting outbreak. According to a farm survey in Sweden, tail biting outbreaks commonly affect only one pen per batch, which restricts tail biting to a maximum of 13 pigs in Sweden compared with a maximum of, e.g., 24 pigs in other countries. Another is that it is easier to identify both the biter and the bitten pigs in a pen with fewer pigs. This enables removal of the biter, which is the most common action, and treatment or removal of bitten pigs [35]. Smaller group size also reduces the need for mixing piglets at weaning. Mixing is stressful for the individual and increases the risk of spread of disease, but a clear association with tail biting has not been found [3].

Swedish farrowing units are typically designed to keep one whole batch of sows (25–50 sows) within the same farrowing room. Thus units for growers commonly hold 300–600 pigs per compartment, although finisher units commonly hold 400 pigs per compartment. One main reason for keeping the whole batch within the same room is high building costs in Sweden, although from an animal health point of view fewer animals per unit would be preferable. Farrowing and nursery units are commonly connected to each other, so piglets only have to walk along a hallway to reach the nursery unit at weaning. This minimises stress compared with systems that involve transportation and also makes it possible to keep litters intact at weaning, to avoid mixing. There is, however, no ban regarding selling pigs directly at weaning. Piglets in Sweden tend not to be sold at weaning, as the animals are sensitive during this period and highly susceptible to pathogens, and therefore do not leave the farm until after the grower period (~30 kg LW).

#### 3.2.3. Flooring

Fully slatted floors are banned in Sweden and 62–75% of the total floor area must consist of solid flooring, calculated depending on live weight as: 0.10+(body weight (kg LW)/167)) [25]. In EU legislation, fully slatted floors are allowed. The regulations for slat width are the same in Sweden and in the EU: 11 mm for piglets, 14 mm for weaners, 18 mm for rearing pigs and 20 mm for gilts after service and sows [13]. In Sweden the rod with is also regulated to 50 mm for piglets and weaners and 80 mm to rearing pigs, gilts and sows. With partly slatted floors the function of the pen fits the needs of the pigs to a greater extent than with fully slatted floors as it leaves the pigs with a more comfortable sleeping area. Moreover, floor type greatly affects the ability to provide pigs with manipulable materials to enable exploratory behaviour, such as straw. Slatted floors are designed to allow urine and faeces to pass through and are therefore not suitable for provision of manipulable material [3]. Pens with partly slatted partly solid flooring or consisting of only solid flooring are compatible with the provision of manipulable material on the solid area.

#### 3.2.4. Air Quality and Light

EU Council Directive 98/58/EC of July 1998 states that “air circulation, dust levels, temperature and relative humidity should be kept within acceptable limits”. However, no specific limits are mentioned there or in later directives. According to Swedish legislation, the relative air humidity may not exceed 80% unless the indoor temperature is below 10 °C [25]. If the indoor temperature falls below 10 °C, the sum of relative air humidity and temperature may not exceed 90%.

Poor air quality has been identified as a risk factor for tail biting [37]. Sweden sets specific maximum levels of different gases within pig facilities, in order to ensure acceptable air quality: ammonia (NH_3_) concentration should not exceed 10 ppm, carbon dioxide (CO_2_) should be maximum 3000 ppm and hydrogen sulphide (H_2_S) maximum 0.5 ppm [25]. These gases are not permitted to exceed the maximum concentration other than temporarily, e.g., when removing slurry [25]. The concentration of organic dust particles may not exceed <10 mg/m^3^ [25]. Concentration of gases are measured by animal welfare inspectors within the official animal welfare control to ensure compliance with the legal requirements [43]. These maximum levels are lower than those accepted in most other EU Member States, if these states have any legal demands other than the EU Pig Directive [42]. Housing environment with high levels of noxious gases is a concern for both pig and human health, as enhancers of respiratory disease. Respiratory disease has been found to be associated with tail biting on farms [44].

According to EU Council Directive 98/58/EC “animals kept in buildings must not be kept in either permanent darkness or without an appropriate period of rest from artificial lighting”. Natural light must be supplemented with artificial light if the natural light is insufficient for the animals’ physiological or ethological needs. Under Swedish legislation, pig houses (unless those in use before 1989) must be provided with windows and pigs must have natural daylight and enough supplementary light to support their daily rhythm and behavioural needs (at least 8 h per day of at least 40 Lux). Daylight hours under Swedish winter conditions are commonly less than 8 h per day around 3 months per year, and supplementary lighting commonly needed. Light is measured at official animal welfare control to ensure compliance with the legal requirements [43]. The associations between light and tail biting in pigs seem not to have been studied. However, it has been found that light only significantly affected few behaviours; lying down inactively was more common in dim light and defecated more in brighter light [45]. Furthermore, they concluded that 40 Lux was not aversive nor strongly preferred by pigs [45].

#### 3.2.5. Manure Handling Systems

The manure handling system does not affect the occurrence of tail biting directly, but it may do so indirectly by allowing/preventing farmers from giving an adequate amount of manipulable material to their pigs. Swedish legislation recommends that farrowing units have manure systems that can handle large amounts of straw [25]. The majority of pig farms in Sweden have liquid manure systems for growing pigs. Ropes/cables with arm scrapers beneath slatted flooring areas is the most common system in farrowing units [46] and in nursery and finisher units [35]. When asked how frequently straw caused blockages or other problems with their manure handling system, in the Swedish survey 56% nursery unit farmers and 81% finishing pig farmers reported that they had never experienced such problems [35]. Within Europe, vacuum systems are commonly used to remove manure [47]. In the Swedish survey, pull-plug vacuum systems were only found in 13% of nursery units and 7% of finisher units [35]. In Swedish farrowing units, vacuum manure systems are occasionally found [46]. If vacuum systems are used, the pipe diameter must be at least 300 mm and the manure must be removed at least every 14 days [25]. To avoid blockage of slats or farther along in the slurry system, chopped straw is commonly used for growing pigs in Sweden [35].

EU Council Directive 95/58/EC requires all animals to be provided with a wholesome and appropriate diet, in sufficient quantities and regular intervals [13]. Feed and water equipment should also minimise the risk of contamination. There are no further regulations or specific requirements in the EU legislation. Under Swedish legislation, feeding and water systems must be dimensioned so that the animals can consume their feed “calmly and in a natural way”. A definition for what can be considered as “natural and calmly” is not provided but this statement is supported by minimum requirements for number of drinkers and length of feeding trough to avoid aggression at feeding. If pigs are fed in a transponder or responder system, all pigs must be able to eat their daily ration within 12 h [25].

#### 3.2.6. Feed and water

Within Sweden, most fattening pigs are fed ad lib., but are put on a restrictive feeding regime after 65 kg LW. Both wet and dry feeding systems are used and are often automated. The average daily gain from weaning until 30 kg LW is 474 g/day, while during the fattening period (30–120 kg LW) pigs grow on average 946 g/day [48]. Inappropriate feeding composition or trouble with feeding equipment is regarded as the most common cause of tail biting outbreaks in Swedish fattening units [35]. Some farmers in Wallgren et al. 2016 reported that a single delay in feeding time can trigger tail biting [35]. This has been confirmed in an epidemiological study, where the occurrence of tail biting was 14 times higher when feeding time was variable [37]. Deviant patterns in feeding behaviour and feed intake have also been shown to predict tail biting outbreaks several weeks in advance [49].

The scientific opinion from EFSA on pig welfare risks associated with tail biting concludes that competition for feed constitutes one of the major risks for tail biting [3]. European Union Council Directive 2008/120/EC states that all pigs must be able to access feed at the same time if fed restrictively on group level, but does not specify how to fulfil this requirement [42]. Swedish legislation clearly specifies the space required along the feeding trough to ensure that all pigs can physically eat at the same time [25]. For growing pigs (30–130 kg LW), the minimum space along the feeding trough is determined as: 0.164 + (body weight (kg LW)/538) [25]. To our knowledge, other EU Member States have no additional specifications on sufficient feeding space [42,50]. For pigs of 30 and 100 kg LW, the Swedish formula corresponds to a trough length of 22 and 35 cm/pig respectively. In a study including data from 233 Swedish fattening units, restricted liquid feeding in troughs <30 cm/pig was found to increase the prevalence of tail biting, while use of feeding troughs >34 cm/pig did not further reduce the level of tail biting [29].

Swedish legislation demands that all pigs have permanent access to a sufficient quantity of fresh water, with at most 40 pigs per water cup and 20 pigs per water nipple when fed dry feed [25]. For groups of over 30 pigs, at least two watering sources must be provided. This requirement is also included in animal welfare legislation in Austria and Germany [42].

### 3.3. Pig Health

It is well known that tail biting can lead to inflammation and internal spread of infection due to bacteria entering the open wound. There are several reports of tail lesions being correlated with other pathological findings such as abscesses, arthritis and lung lesions, in slaughter pigs [51,52,53]. Evidence of reduced health as a predisposing factor to tail biting is less frequent, although general pig health status on the farm is believed to be related to the risk of tail biting [44,54]. In a Finnish survey of farmers on the efficiency of different preventive measures among undocked pigs, “taking care of animal health” was rated very important [55]. In a similar study in the Netherlands, conventional Dutch pig farmers ranked “suboptimal health” as one of the top three most important risk factors for tail biting [6].

Sweden is declared free from Aujeszky’s disease [56] and PRRS [57] and national surveillance of salmonellosis and swine dysentery show that occurrence of these diseases are very low [58]. Typical vaccination programs includes vaccination against porcine circovirus type 2 (PCV2), *Mycoplasma hyopneumoniae,* and in some farms also against *Actinobacillus pleuropneumoniae* (App) and/or *Lawsonia intracellularis.* In a detailed comparison of antimicrobial usage in farrow-to-finish herds in four EU-countries, Sweden had the lowest treatment incidence (TI) based on Defined Daily Doses Animal per 1000 pig-days at risk, while German herds had the highest overall use [59]. The largest difference was found in weaners, where Swedish farms reported a median TI of 6.1 treatments per 1000 pig-days while farms in Belgium, France and Germany reported a median TI of 339.5, 320.1 and 487.6 treatments [59]. A recent compilation of medical health data on 147 Swedish fattening pig herds and 73 piglet-producing herds found that for suckling piglets, the median number of antibiotic-treated cases per 1000 live-born piglets was 245 (interquartile range, (IQR): 120–358), while for weaners the median was 78 cases per 1000 weaned pigs (IQR: 34–166), and in finishers it was 46 antibiotic-treated cases per 1000 slaughtered pigs (IQR: 24–98) [60,61]. Mortality post weaning and during the fattening period is also low in Sweden compared with many other countries [62]. Data from the Swedish national production database show an average mortality of 2% from weaning to 30 kg and of 1.8% during the fattening period [48]. Altogether, this indicates high health status on Swedish pig farms, which may be one important factor in Sweden’s success in rearing pigs with intact tails. Since several risk factors for tail biting also influence health, it is likely that taking action to prevent tail biting (i.e., reducing stocking density and avoid mixing of pigs), will also improve the general health status on the farm.

One contributing factor to the high health status on Swedish pig farms may be the generally high weaning age. Under Swedish legislation, piglets may not be weaned before 4 weeks of age. However, to ensure that no piglets in the batch are less than 28 days at weaning, most Swedish farmers tend to wean their piglets at approximately 5 weeks of age (33.1 days on average) [48]. New regulations permit a maximum of 10% of piglets in a batch to be weaned at <26 days if the farm is associated with a specific health scheme which also takes into account that there are no behavioural deviations after weaning such as tail biting [25]. Within other European Member States, piglets are commonly weaned before 28 days of age.

Biosecurity status among Swedish pig farms have been found to be somewhat higher than in Belgium and France but variation between farms is large and there is room for improvement of the level of biosecurity in many herds [63].

### 3.4. Environmental Enrichment

One of the highest risk factors for tail biting is lack of long straw [3]. Straw increases explorative behaviour and reduces behaviours such as tail biting [3]. According to EU Council Directive 2008/120/EC, “pigs shall have permanent access to a sufficient amount of material to enable proper investigation and manipulation activities, e.g., straw”. Under Swedish legislation, farmers are obliged to provide pigs with litter material that meets their exploratory and comfort needs, in terms of both characteristics and quantity [25]. Neither European nor Swedish legislation suggests how to assess whether this criterion has been met.

In a survey all Swedish farmers reported that they provided their pigs with some sort of manipulable material and that 99% provided straw [35]. Considering partly slatted flooring systems, the amount of straw was around 29 g/pig/day for growers (range 8–85 g/pig/day, *n* = 29) and 50 g/pig/day for finishers (range 9–225 g/pig/day, *n* = 22). Swedish pig farms are predominantly located in areas with much grain production and hence also good straw availability, which is probably why straw is the most commonly manipulable material provided. Farms with a higher straw ration observed tail biting in their production less often [35]. Increased straw ration also increases straw-directed behaviours, while decreasing pen-directed behaviours and occurrence of tail lesions [38]. Few farmers (24%) in the survey wanted to increase the straw ration beyond the minimum requirement, mainly due to concerns regarding pen hygiene and manure handling, while problems with manure handling systems were uncommon [35]. Later studies on commercial Swedish production have shown that pig and pen hygiene is good, irrespective of the straw ration [64,65].

In organic pig production, growing/finishing pigs provided with silage in addition to straw have been shown to utilise nutrients in the silage and to respond with a milder reaction to social interactions than pigs only provided with straw, and thus had fewer wounds from adverse social interactions [66]. Therefore, provision of silage in an environment enriched with straw can further improve pig welfare. Pigs fed silage during the latter part of the finishing period in a subsequent study were found to perform more feed-directed behaviours and less behaviours directed towards other pigs and pen fittings [67].

According to the survey of Swedish farmers [35], they commonly provide their pigs with straw once daily (76.5% of grower farms, 82.9% of finisher farms), but the rate ranged from twice daily to every second week for partly slatted systems. Fresh provision of straw is important to stimulate exploratory behaviour. Straw manipulation is highest one hour after straw allocation and newly provided straw seems particularly interesting to pigs [68,69,70,71]. Previous studies have shown that providing straw several times a day has little impact on straw manipulation and redirected behaviour [72]. However, the way of providing straw seems to affect overall activity biting [73,74]. Straw is commonly provided manually, although some Swedish farms have automatic systems for straw provision. Providing straw manually takes time, but gives the farmer the opportunity to perform an ocular check of the absence of mold in the bales, as well as, to interact with and observe their animals. Healthy pigs will immediately get up and start to explore the straw and sick or lame pigs can therefore be spotted more easily.

Using straw has been raised concerns regarding the hygienic hazards. However, a German study of the hygienic status of e.g., straw with special emphasis on pathogenic bacteria, concluded that straw was not likely to pose a hygienic risk to pigs and are suitable as enrichment material [75].It is possible to use money from Compensation for extra animal care for sows provided by the Swedish government to test e.g., straw for deoxynivalenol (DON) and zearaleon (ZEN) at the Swedish veterinary institute [76]. Furthermore, Sweden does not have e.g., African Swine Fever (ASF) and Aujeszky’s disease (AD), and Swedish veterinary institute recommended not to import straw from other EU member states [77]. Therefore, the Swedish veterinary institute recommend straw as a safe litter material [78].

## 4. General Discussion

Sweden has a long tradition in rearing undocked pigs, with tail biting considered to be a minor problem on most Swedish farms [35]. Housing conditions and management in Sweden differ in many aspects from those in other EU Member States, partly because the requirements are higher in Swedish animal welfare regulations than in EU legislation and partly because of tradition.

Tail biting is caused by multiple factors and thus requires multifactorial solutions. The specific requirements imposed by Swedish animal welfare legislation eliminate some known risk factors for tail biting. These requirements, which include lower stocking density, provision of sufficient feeding space, a ban on fully slatted flooring and maximum permissible levels for noxious gases, are crucial for Swedish farmers’ success in rearing pigs with intact tails. Another requirement, for regular provision of an adequate amount of litter material sufficient to fulfil the pigs’ need for performing exploratory behaviour, is highly important, but all other factors also need to be in place. Rearing undocked finisher pigs to 110 kg LW on 0.65 m^2^ of fully slatted flooring, as under the current EU legislation, is unlikely to eliminate tail biting, even if straw is provided. The limited feeding space provided in EU nursery units may also increase tail biting. Swedish herd health veterinarians report that tail biting increases when grower pens become too crowded, e.g., when more piglets than expected are weaned at the same time or when the farmers are unable to move finisher pigs to a fattening unit in time (Farm and Animal Health, unpublished data). Lack of feeding space is considered to be the main cause of tail biting outbreaks in these situations. To prevent tail biting and reduce the need for tail docking in future, we strongly recommend that current EU legislation be revised to better match the biological needs of pigs. In our view, providing materials such as chains or wood logs, which are more compatible with fully slatted floorings but do not solely possess important characteristics such as changeability and edibility and cannot be manipulated by several pigs at the same time, does not fully comply with current legislation. This position has also been taken by the EU Commission [16]. Such materials will not fulfil the pigs’ behavioural needs and therefore will probably not enable rearing of pigs with intact tails. Pen design needs to be modified to enable provision of suitable rooting material, and fully slatted floors are not appropriate in their current form. It is not only the housing conditions and legal requirements that affects the upcoming of tail biting, but also the management, Swedish farmers do not only manage tail biting through prevention, but also handles tail biting when outbreaks occurs [35]. When tail biting is seen, the pigs are commonly provided with extra straw or other toys. Further, identification and removal of the biter and bitten pig to a separate hospital pen, which is compulsory in all pig compartments in Sweden [25], and reintroduction of the pig is performed if possible. Identification of the biting animal is eased by the commonly small groups of pigs and outbreaks commonly affect one pen per outbreak. This indicates that Swedish farmers are also good in managing and preventing tail biting outbreaks from developing.

The housing and management requirements in Swedish animal welfare legislation make production more costly [62]. Furthermore, the harsh Swedish winter climate results in high heating demand and requires proper insulation capacity, factors that increase the building and maintenance costs. On the other hand, the performance of Swedish fattening pigs exceeds that in most other countries. The latest InterPIG-report, comparing production results from the major pork producing countries in the EU and Brazil, Canada and the US, shows that Sweden has the highest growth rate and a low feed conversion rate from weaning until slaughter [62]. In 2018, the average weight gain was 946 g/day/pig during the finisher period (30–120 kg LW), and the energy needed for growth was 25.4 MJ NE (net energy) per kg weight gain according to the Swedish national production database [62]. The 25% best farms had an average weight gain of 1027 g/day. Less feed needed for growth means money savings. Within finisher production, feed costs account for about 40% of the variable costs [62].

The InterPIG-report [62] does not consider the fact that Swedish pigs (and all pigs in EU) are reared entirely without the provision of antibiotics for preventative purposes, which is common in, e.g., the U.S. This means that Swedish pig production is even more efficient than the report shows. High growth rates increase stress on pigs, increasing the risk of tail biting if the pigs do not receive the feed required to maintain this high growth. Swedish farmers have found that incorrect feed composition or a too-restricted feeding regime can lead to problems with tail biting.

In parallel with high growth rates, within Swedish pig production general health is high and mortality low during the weaner and fattening periods, which also leads to reduced economic losses. Among countries monitored in the most recent InterPIG-report, mortality from weaning to slaughter was 5.8% on average [62]. US had the highest mortality (8.6%), followed by Spain (7.1%), while Sweden had the lowest mortality after weaning (3.6%). Therefore, although not as many piglets are weaned in Sweden as in the US and Spain (due to minimum weaning age being higher), the number of pigs finished per sow per year is higher in Sweden [62].

In genetic selection for reduced tail biting behaviours, tracing biters with a known pedigree (from nucleus down to fatteners) back to the breeding organisation could be a useful tool. In such an approach, all piglets in a litter would be given a litter identity so that any negative behaviour could be traced back to relatives and selected out. Selection would need to be performed in the environments countered by production animals, and thus the future environment and environmental aspects relevant for tail biting (which currently differ between, e.g., Sweden and many other countries) would need to be defined.

There is a growing concern among modern consumers about farm animal welfare [79]. In order to maintain the good reputation of EU pork production, consumer demands must be considered, and sustainable production must be enabled. Rearing conditions that lead to behavioural problems, e.g., tail biting, which require routine mutilation of pigs, such as tail docking, are not in line with the animal welfare concerns among consumers. For EU pig production to become a sustainable part of future food production, all aspects of sustainability, i.e., social, ecological and economic, must be addressed.

One of the major challenges with future prevention of tail biting is to solve the multifactorial complexity of the problem. There are no quick fixes or single solutions (e.g., more straw or larger space allowance). The Swedish example shows that all important factors need to be considered simultaneously. It is likely that the more detailed regulations on, e.g., ventilation in Swedish legislation play an important role in the overall environment of the pigs, and hence also in their ability to cope with stress. Even if all external resources are provided, internal factors such as genetics may increase the risks of tail biting.

Another major step in transforming from a docked to an undocked pig production is implementation of the EU Pig Directive. The pigs produced in the EU are still mainly docked and the EU still has to ensure compliance by Member States with the Directive. In order to achieve full compliance, considerable changes in current production systems are needed, requiring investment and probably increasing production costs. Hence, the whole EU must act together to avoid compromising pig production by individual farmers or Member Countries.

## 5. Conclusions

Sweden has a long tradition in rearing undocked pigs and yet tail biting is a minor problem within Swedish pig production. This review indicates that the requirements within Swedish animal welfare regulation for lower stocking density, provision of sufficient feeding space, no fully slatted flooring, specified maximum levels of noxious gases and regular provision of adequate amounts of litter material sufficient to enable pigs to performing exploratory behaviour are crucial for the success of Swedish pig producers in rearing pigs with intact tails. To prevent tail biting and reduce the need for tail docking in the future, we strongly recommend that current EU legislation be revised to more clearly match the biological needs of pigs, as is done in Swedish legislation.

## Figures and Tables

**Figure 1 animals-09-00812-f001:**
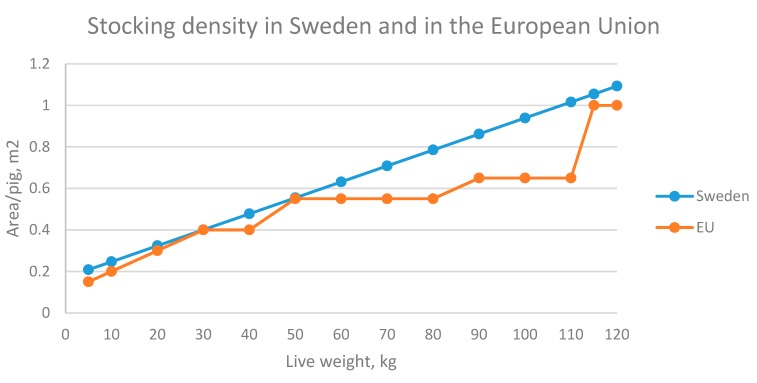
Minimum stocking density requirement for growing pigs in the European Union (EU) and in Sweden. Note that weaning weight in Sweden is ~10 kg live weight (LW) and pigs <10 kg LW are commonly still housed with their dams in the farrowing pen, while weaning weight in the EU is <10 kg LW (due to lower weaning age) and hence piglets are moved to grower pens at lower LW.

**Table 1 animals-09-00812-t001:** Management routine in pig production in EU member states solely complying with the Council Directive 2008/120/EC [13] and Sweden Respectively.

Management Routine	Legislation Applied	
	EU	Sweden
Tail docking	No (but still practiced)	No
Crating of Sows	4 weeks during lactation and 4 weeks after insemination	No
Straw provision	No	Yes
Weaning age	28 days (21 days if they are moved into specialised housings	28 days (a few piglets per batch may be weaned at 21 days, if part of a special control program)
Space allowance sow and piglets	2.25 m^2^ 0.3–1 m^2^ dependent on weight See Figure 1	6 m^2^
Space allowance growing finishing pigs	0.3–1 m^2^ dependent on weight See Figure 1	= 0.17 + (kg LW)/130. See Figure 1
Slatted floors	Fully slatted floors	Partly slatted floors (maximum slatted floors ~35%)

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
