# Peer review of "Rearing Pigs with Intact Tails—Experiences and Practical Solutions in Sweden"

_animals, 2019, doi:10.3390/ani9100812_

Round 1

Reviewer 1 Report

The manuscript is very interesting, well written and complete.
Personally I welcome the fact that the authors have attached importance to several factors responsible for the onset of tail biting without focusing predominantly on environmental enrichment as many do. This is also the position of my research group.
Here some minor comments:

Line 68: to lie down at the same time

Line 116. I suggest “the most commonly suspected cause of tail biting reported by grower farmer was…”

Line 121: delete (5%)

Line 210: it is probably more appropriate to state that Sweden has specific demands for air quality in the pig facilities, whereas most other EU countries do not.

Lines 213- 220: a lighting period lasting a least 8h at 40 lux is the minimum level required also by current EU legislation thus this aspect cannot be considered a prerogative of Swedish farms. Did the authors wish to refer to the Swedish obligation to provide natural lighting?
As concerns a possible negative association between high lighting regimes and injurious behaviours, Italian researches on heavy pigs have demonstrated that animals  receiving higher illumination levels (both in terms of light duration and light intensity) were calmer than pigs raised according to the minimum mandatory levels for light duration and intensity ( Martelli et al., 2010 Berl. Munch. Tierarztl. 123(11-12), pp. 457-462 ; Martelli et al., 2015, J. Anim. Sci. 93(2), pp. 758-766 ). Therefore it would not seem that what has been observed in poultry by Danish researchers (severe pecking at 30 lux and gentle pecking at 3 lux) is fully applicable to swine.

Line 262: watering instead of feeding?

Lines 280-282: please provide some references concerning the comparison between Sweden and the rest of EU.

Lines 294-295: please check this statement (I am sorry but I am not able to find it in the EFSA report). Also according to EU legislation, the minimum age for weaning is of 28d. It is true that when batch-farrowing is applied then the weaning age may be reduced but this is only a slight reduction (about 26 d). To my best knowledge (see also AHDB statistics 2016) early weaning at 21 days is not a so frequent practice in the EU and it is applicable, according to legislation, only if (expensive) specialised housings systems are available.

Lines 304-305: the survey was carried out by telephone. Taking into account the well-known “social desirability bias” which can affect an oral response, author should clearly specify that 99% of Swedish farmers declared of providing pigs with straw.

Lines 354-357: this is also the EU view as clearly reported in  COMMISSION RECOMMENDATION (EU) 2016/336

Line 409: It is highly questionable that EU has made only a little effort. Surely the result is not satisfactory at present but this fact must not diminish the extent of the effort itself. I think it would be more correct to say that extensive work has already been done but certainly there is much more to do.

Author Response

Reviewer 1

Comments and Suggestions for Authors

The manuscript is very interesting, well written and complete.
Personally I welcome the fact that the authors have attached importance to several factors responsible for the onset of tail biting without focusing predominantly on environmental enrichment as many do. This is also the position of my research group.
Here some minor comments:

Line 68: to lie down at the same time -Modified as suggested

Line 116. I suggest “the most commonly suspected cause of tail biting reported by grower farmer was…” -Modified as suggested

Line 121: delete (5%) -Modified as suggested

Line 210: it is probably more appropriate to state that Sweden has specific demands for air quality in the pig facilities, whereas most other EU countries do not. -Modified as suggested

Lines 213- 220: a lighting period lasting a least 8h at 40 lux is the minimum level required also by current EU legislation thus this aspect cannot be considered a prerogative of Swedish farms. Did the authors wish to refer to the Swedish obligation to provide natural lighting?
As concerns a possible negative association between high lighting regimes and injurious behaviours, Italian researches on heavy pigs have demonstrated that animals  receiving higher illumination levels (both in terms of light duration and light intensity) were calmer than pigs raised according to the minimum mandatory levels for light duration and intensity ( Martelli et al., 2010 Berl. Munch. Tierarztl. 123(11-12), pp. 457-462 ; Martelli et al., 2015, J. Anim. Sci. 93(2), pp. 758-766 ). Therefore it would not seem that what has been observed in poultry by Danish researchers (severe pecking at 30 lux and gentle pecking at 3 lux) is fully applicable to swine.

Text has been modified. Poultry reference has been deleted and new references added. However, we have not extended the discussion.

Line 262: watering instead of feeding? -Modified as suggested

Lines 280-282: please provide some references concerning the comparison between Sweden and the rest of EU.

Text has been added

Lines 294-295: please check this statement (I am sorry but I am not able to find it in the EFSA report). Also according to EU legislation, the minimum age for weaning is of 28d. It is true that when batch-farrowing is applied then the weaning age may be reduced but this is only a slight reduction (about 26 d). To my best knowledge (see also AHDB statistics 2016) early weaning at 21 days is not a so frequent practice in the EU and it is applicable, according to legislation, only if (expensive) specialised housings systems are available.

Statement has been changed to “piglets are commonly weaned before 28 days of age” instead and other references has been added to support this statement.

Lines 304-305: the survey was carried out by telephone. Taking into account the well-known “social desirability bias” which can affect an oral response, author should clearly specify that 99% of Swedish farmers declared of providing pigs with straw. -Modified as suggested

Lines 354-357: this is also the EU view as clearly reported in  COMMISSION RECOMMENDATION (EU) 2016/336 – text added

Line 409: It is highly questionable that EU has made only a little effort. Surely the result is not satisfactory at present but this fact must not diminish the extent of the effort itself. I think it would be more correct to say that extensive work has already been done but certainly there is much more to do. -Modified as suggested

Reviewer 2 Report

The review paper is concerned with a topic of high importance in EU pig industry: rearing pigs with intact tails. The authors clearly summarise knowledge from Swedish production of undocked pigs and provides an overview of hands-on practical solutions that can facilitate other Members States to ensure compliance with the EU Directive. The manuscript is well-written and the general discussion provides insights of practical interest. In general, the manuscript is worthy of publication and I have only few minor corrections to suggest as listed below:

1. P1, L 3-4: Part of the front matter (i.e. Author list and their affiliations) is missing. Please, include this info as for the Journal guidelines.

2. P1, L 8: Simple Summary - Please, consider to add ‘can’ between ‘biting’ and ‘be’.

3. P3, L126-127: Please, provide few references.

4. P3, L 132: I suppose something is missing in this sentence. Perhaps, ‘more/less likely’ or similar? I couldn’t check ref. 42 because the list of references was missing.

5. P3-6, L 138: I would suggest to divide paragraph 3.2 into further sub-headings (e.g. Stocking density, Husbandry and general management, Flooring system, Air quality etc.) so that the different external factors can be easily identified by the readers.

6. P 6, L 237: I think that naming the study (e.g. xx et al. reported that) would make this sentence sounds better than ‘that study reported’.

7. P 6, L 253: Do the authors mean ‘water sources’ instead of ‘feeding sources’? If not, would this sentence (‘For groups…be provided’) sound better into the previous section (L 242-250) when the authors talk about feeding requirements?

8. P 8, L 341-343: Please, remove one of the two ‘that’ at the beginning of the sentence and consider to provide a reference to support the citation.

9. P 8, L 368: Please, consider to move ‘a’ before ‘too’.

10. P9, L 421: Part of the back matter (i.e. References and Acknowledgments) is missing. Please, include this info as for the Journal guidelines.

11. P9, L 413: Supplementary material - The supplementary files (Figure S1, Table S1 and Video S1) were neither available as part of the peer-review process nor included into the main text to their first citation as for normal figures/tables. I recommend the authors to upload a new version of these documents.

Author Response

Reviewer 2

Comments and Suggestions for Authors

The review paper is concerned with a topic of high importance in EU pig industry: rearing pigs with intact tails. The authors clearly summarise knowledge from Swedish production of undocked pigs and provides an overview of hands-on practical solutions that can facilitate other Members States to ensure compliance with the EU Directive. The manuscript is well-written and the general discussion provides insights of practical interest. In general, the manuscript is worthy of publication and I have only few minor corrections to suggest as listed below:

P1, L 3-4: Part of the front matter (i.e. Author list and their affiliations) is missing. Please, include this info as for the Journal guidelines. -Modified as suggested P1, L 8: Simple Summary - Please, consider to add ‘can’ between ‘biting’ and ‘be’. -Modified as suggested P3, L126-127: Please, provide few references.- Answer: We have searched for, but not found reports from such studies. This was just speculation from our side thus we have now deleted this part. P3, L 132: I suppose something is missing in this sentence. Perhaps, ‘more/less likely’ or similar? I couldn’t check ref. 42 because the list of references was missing. -Modified as suggested P3-6, L 138: I would suggest to divide paragraph 3.2 into further sub-headings (e.g. Stocking density, Husbandry and general management, Flooring system, Air quality etc.) so that the different external factors can be easily identified by the readers. - Subheadings now added. P 6, L 237: I think that naming the study (e.g. xx et al. reported that) would make this sentence sounds better than ‘that study reported’. -Modified as suggested P 6, L 253: Do the authors mean ‘water sources’ instead of ‘feeding sources’? If not, would this sentence (‘For groups be provided’) sound better into the previous section (L 242-250) when the authors talk about feeding requirements? -Modified as suggested P 8, L 341-343: Please, remove one of the two ‘that’ at the beginning of the sentence and consider to provide a reference to support the citation. -Modified as suggested P 8, L 368: Please, consider to move ‘a’ before ‘too’. -Modified as suggested P9, L 421: Part of the back matter (i.e. References and Acknowledgments) is missing. Please, include this info as for the Journal guidelines. -Modified as suggested P9, L 413: Supplementary material - The supplementary files (Figure S1, Table S1 and Video S1) were neither available as part of the peer-review process nor included into the main text to their first citation as for normal figures/tables. I recommend the authors to upload a new version of these documents. – Reviewer seems to have been reading another print out of manuscript. However, minor modification has been performed as suggested. We did not have any supplement.

Reviewer 3 Report

General comments:

Good review, however for it to support the intention of changing farming practices (underline the importance of successful management practices in Sweden compared to other EU countries), the economical aspects should be added. To bring different countries with very different legislations and ways to approach housing, management and preventive strategies at individual/group levels, one should be able to show the benefits of using particular solutions: are there studies showing growth rate/feed consumed across different space allowances/tail biting prevalences? By showing the marginal wins for different farming approaches one could also motivate farmers/consumers for higher welfare standard as pig production remains highly competitive.

Specific comments:

63-70 Due to the descriptive and comparative nature of the article, it will be beneficial to provide a table comparing the main husbandry parameters between Sweden and EU Member State countries. 124-130 If this to be used at EU level, either reference around castration practices in non-Scandinavian countries or some sort of additional comparison should be provided. 148- As mentioned previously, the text will benefit if these numbers will be presented in the form of a table (even considering the presence of figure 1) 190 What was the reason for this ban? While comparing practices leading to a common denominator (tail biting events) it should be possible to draw parallels between management decisions and/or building parameters affecting animals. 238-239 “naturally and calmly” is that a definition from legislation, or how do authors define these two terms under the feeding scenario? 280 How is mortality rates compared to the number of piglets born (Sweden vs EU countries) 282 Are there any numbers showing the main causes of mortality in different age groups across different countries?

Author Response

Reviewer 3

Comments and Suggestions for Authors

General comments:

Good review, however for it to support the intention of changing farming practices (underline the importance of successful management practices in Sweden compared to other EU countries), the economical aspects should be added. To bring different countries with very different legislations and ways to approach housing, management and preventive strategies at individual/group levels, one should be able to show the benefits of using particular solutions: are there studies showing growth rate/feed consumed across different space allowances/tail biting prevalences? By showing the marginal wins for different farming approaches one could also motivate farmers/consumers for higher welfare standard as pig production remains highly competitive.

The Authors agree that economic aspects are relevant. Some major aspects such as effects on growth etc. are included in the manuscript. Economy is a wide concept though and the reviewer probably refer more specifically to the financial aspects of economy (i.e. when resource use is estimated in monetary units). The authors agree that that would be a very interesting paper, however, this paper is about the production and animal welfare aspects of tail biting, providing input for such a financial simulation where e.g. net profit/win can be estimated. Such a paper, comparing financial outcome between countries need detailed information on pig production from all countries/production systems included. This paper, however, presents results and experiences from the Swedish system, showing that it is biologically possible to rear pigs with intact tails without severe tail biting issues. Thus it is possible to follow the current EU-legislation.

Specific comments:

63-70 Due to the descriptive and comparative nature of the article, it will be beneficial to provide a table comparing the main husbandry parameters between Sweden and EU Member State countries.

Table has been inserted.

124-130 If this to be used at EU level, either reference around castration practices in non-Scandinavian countries or some sort of additional comparison should be provided.

Answer: Castration was not a main point here, thus this part has now been deleted.

148- As mentioned previously, the text will benefit if these numbers will be presented in the form of a table (even considering the presence of figure 1)

Table has been inserted.

190 What was the reason for this ban? While comparing practices leading to a common denominator (tail biting events) it should be possible to draw parallels between management decisions and/or building parameters affecting animals.

Answer: Now clarified in LM 197-200 “There is however no ban regarding selling pigs directly at weaning. Piglets in Sweden tend not to be sold at weaning, as the animals are sensitive during this period and highly susceptible to pathogens, and therefore do not leave the farm until after the grower period (~30kg LW).”

238-239 “naturally and calmly” is that a definition from legislation, or how do authors define these two terms under the feeding scenario?

Answer: Exactly these words are stated in the Swedish legislation but no definition is provided. We have clarified this in the text.

280 How is mortality rates compared to the number of piglets born (Sweden vs EU countries) –

Answer: Pre-weaning mortality rates are somewhat higher in Sweden compared to other countries but then we also in general have a longer suckling period. Since this paper does not discuss housing or management of sows and suckling piglets we choose not to add this information.

The word “Altogether” added in Line 320 to clarify that everything discussed in this section sums up to the conclusion about high health status, not only the low mortality.

282 Are there any numbers showing the main causes of mortality in different age groups across different countries? –

Answer: We have searched but not found any reports on this for finishers and weaners. Data on causes for mortality are available for sows and suckling piglets but we don´t see the point in adding that information in this paper. 

Reviewer 4 Report

Wallgren and colleagues presented descriptive results on tail biting in Sweden. In general, tail biting issue is a really hot topic and some results are of interest. However, too many data are provided, I would suggest a  reduction in order to focus on “high-quality” information only. Most relevant aspects need to be discussed first, a clearer chapter organization needs to be made (the "general discussion" is in some parts a repetition of previously discussed parts).

I also suggest some specific clarifications and language amendments:

Line 14: environment

Line 16: “EU regulations” is not correct, as it is one directive

Line 17: can be?

Line 24: “better harmonised with the biological needs..” what does it mean?

Line 25: ..it is (mainly) an indicator of.. I would add “mainly” or something else, as tail biting is still reported (rarely) in extensive outdoor production

In general, simple summary it is very similar to the abstract, I would suggest to better diversify them.

Introduction: I would suggest revise the references to this part as it is not clear on which basis they have been chosen. Eg line 42: ref 1-4? They are neither the most recent nor the most indicative to support the general statement.

Line 48: “only a few countries” ..with a minor pig production in comparison to other EU countries 

Line 52: the directive properly “requires”, not “recommend”. Rather, the 2016 EU Recommendation should also be cited.

Line 57: ref 19 does not fit well with the sentence. I would suggest some primary reference supporting the statement.

Line 58-62: this paragraph can be shortened by citing only the second piece of law. Rather, be more specific in the details of the following paragraph. Ref 22 is in Swedish so please clarify better the data (how much rare? According to a national survery?)

Line 67: more generous space.. how much exactly? (eg xx% more than what requested by EU Directive..).

Line 79: how can be difficult to differentiate bitten and docked tails, given that in Sweden it is forbidden? Do you mean emergency tail docking (so only in case of need as a surgical intervention carried out by a veterinarian with analgesia and anaesthesia?). Is necrosis a common issue in Sweden?

Line 85: please clarify the source of the reference (national guidelines? Official document?)

Line 87: why did you include Norway, while you are discussing Swedish results (see the title)

Line 91: please better clarify the source of the unpublished data.

Line 92: is tail docking banned also in the UK?

Line 99: is it possible to split the percentages of injured and shortened tails? Rather, reporting the difference between the two abattoirs it is not so important.

Line 100-101: sentence not clear

Line 110: do you mean “at least one case? At least one “outbreak”?

Line 118-121: you should also include a comment on the reliability of other emergency measures to manage TB outbreaks. E.g. it is allowed in Sweden of emergency tail docking and how? (surgically, rubber ring etc..). I think that these details are of high interest for countries that are approaching undocked pigs only recently.

Line 125: castrates (better: “barrows”). Comment on behavioural difference in barrows and females should be better supported by a ref on an ethological study. Are pigs bred in mixed or single gender pens in Sweden?

Line 131: please clarify the age at risk. Heavy production (Spain, Italy) are not at high risk.

Line 148-150: no need to report the directive table (and if you choose to do, please revise the intervals of square meters).

Line 151: this is a very interesting point. Please better explain how it is measured in Sweden (automated weighing of animals?) Also clarify how can the data in tab 1 be achieved: I assume that you put 10-13 pig into a pen after weaning and they remain until slaughtering (while in other countries they are moved and regrouped a few times). So, how can be that the more the weight increase the more the area per pig increase? It seems like the pen is progressively enlarged in some way… I disagree with the representation of EU trend: the space is progressively decreasing, given that small pigs are put into a large pen where they grow; once they are near the law limit they are moved into another larger pen, or the group is split.  

Line 199: revise the date

Line 204-212: explain how in Sweden the gas threshold are verified (have farmers some device for continuous gas monitoring? Do official veterinarians use some instrument for measurement?)

Line 210: refer more clearly to Dir. 2008/120/EC

Line 213: correct the date

Line 220: clarify the connection between light and pig behaviour, avoid comparisons with hens and explain how in Sweden the compliance to 40 lux is verified

Line 255: consider EFSA guidelines

Lines 274-280: such data on AMU are poorly informative, you should use dose-based methods (eg. DDDvet). Also the reference to EMA on veterinary sales is debatable (it includes sales for all livestock), you should use dose-based data on pigs specifically in other countries.

A discussion on pig health in Sweden should also consider vaccination plans, level of biosecurity, and major pathogens occurring. Whether it is impossible to discuss all these aspect, they should at least be mentioned.

Line 316: I think in the meanwhile you published as conference proceedings.

Line 324-334: a current concern is the risk of pathogens in straw (eg. considering the risk for ASF, AD, etc.). Do you have in Sweden any procedure to assure a pathogen-free straw?

Line 335: there is a switch from point 3 to point 5

Line 355: recent publications rehabilitated the suitability of chains, although they need to be combined with other enrichment. Consider also the EU Working document suggesting a combination of suboptimal materials

Line 374: growth promoters are banned in all EU states.

Line 419: you have however a level of damage and the reader still may ask which is the treashold of acceptability. Moreover, it seems that most of the achievement is due to prevention, while scientific reports (eg. EFSA) also push on the importance of managing the problem (education of farmers, proper management procedures, number of sickbay areas per farm,…)

Line 421: same unclear sentence than in the abstract.. are you suggesting that in other EU countries the legislation is applied like in Sweden or that the legislation is revised in order guarantee higher welfare standards (given that national Swedish legislation seems much more restrictive in some points)?

Author Response

Reviewer 4¤

Comments and Suggestions for Authors

Wallgren and colleagues presented descriptive results on tail biting in Sweden. In general, tail biting issue is a really hot topic and some results are of interest. However, too many data are provided, I would suggest a  reduction in order to focus on “high-quality” information only. Most relevant aspects need to be discussed first, a clearer chapter organization needs to be made (the "general discussion" is in some parts a repetition of previously discussed parts).

– We see the point to focus on “high-quality” information only, however a lot of statistics and other information from practical pig farming in Sweden is not available in international peer-reviewed paper. We think that the information from sources that may not be consider as High quality can be valuable to an international audience, although the info should be present with caution.

We have reorganised the paper and sectionalised the manuscript to enhance reading.

I also suggest some specific clarifications and language amendments:

Line 14: environment -Modified as suggested

Line 16: “EU regulations” is not correct, as it is one directive -Modified as suggested

Line 17: can be? -Modified as suggested

Line 24: “better harmonised with the biological needs..” what does it mean?

Now clarified in LN 392-396, and hopeful in related parts of the text.

Line 25: ..it is (mainly) an indicator of.. I would add “mainly” or something else, as tail biting is still reported (rarely) in extensive outdoor production -Modified as suggested

In general, simple summary it is very similar to the abstract, I would suggest to better diversify them.

Some changes have been made, but we hope that some similarities in the 2 abstracts would be accepted.

Introduction: I would suggest revise the references to this part as it is not clear on which basis they have been chosen. Eg line 42: ref 1-4? They are neither the most recent nor the most indicative to support the general statement.

Revised

Line 48: “only a few countries” ..with a minor pig production in comparison to other EU countries  -Modified as suggested. That is correct, but the Swedish example still shows that rearing of pigs with intact tails is possible.

Line 52: the directive properly “requires”, not “recommend”. Rather, the 2016 EU Recommendation should also be cited. -Modified as suggested, and reference has been added.

Line 57: ref 19 does not fit well with the sentence. I would suggest some primary reference supporting the statement. -Modified as suggested, and changed into ref 5.

Line 58-62: this paragraph can be shortened by citing only the second piece of law. Rather, be more specific in the details of the following paragraph. Ref 22 is in Swedish so please clarify better the data (how much rare? According to a national survery?)

Text has been modified. Swedish reference is a popular descript of the history of animal welfare in Sweden by a Swedish vet and professor in animal hygiene, that have been active in the area since the 1940ies, and still is(!). In the 1940ies few survey of anything in animal production was performed.

Line 67: more generous space.. how much exactly? (eg xx% more than what requested by EU Directive..).

Information has been added in the new figure 1.

Line 79: how can be difficult to differentiate bitten and docked tails, given that in Sweden it is forbidden? Do you mean emergency tail docking (so only in case of need as a surgical intervention carried out by a veterinarian with analgesia and anaesthesia?). Is necrosis a common issue in Sweden?

This was meant as a general comment on reasons for tail damage, not specifically for Sweden. And No, no tail docked pigs appear at Swedish slaughterhouses. If tails are gone they have been bitten of. This is now deleted from the text.

Line 85: please clarify the source of the reference (national guidelines? Official document?)

Now clarified with translation in the reference.

Line 87: why did you include Norway, while you are discussing Swedish results (see the title)

We want to compare the Swedish figures with what has been seen in other countries, and the official figures from Norway is the best data to compare the Swedish figures with as both countries have routine recordings at slaughter.

Line 91: please better clarify the source of the unpublished data.

Answer: Now specified in LM 93-96.

Line 92: is tail docking banned also in the UK?

No, but some farms do not tail dock, these figures are based on data from those animals

Line 99: is it possible to split the percentages of injured and shortened tails? Rather, reporting the difference between the two abattoirs it is not so important.

Answer: The aim is not to compare the two slaughterhouses but to show that the prevalences of different severity levels of tail injury is about the same in both slaughter houses. Need to be considered together with the next sentence.

Line 100-101: sentence not clear

Now rephrased in LN 105-106 to make the connection to the previous sentence clearer.

Line 110: do you mean “at least one case? At least one “outbreak”?

Answer: yes, this is now clarified in LN 115-117

Line 118-121: you should also include a comment on the reliability of other emergency measures to manage TB outbreaks. E.g. it is allowed in Sweden of emergency tail docking and how? (surgically, rubber ring etc..). I think that these details are of high interest for countries that are approaching undocked pigs only recently.

Emergency docking is not practiced in Sweden. This is now clarified in LN 126-127

Line 125: castrates (better: “barrows”). Comment on behavioural difference in barrows and females should be better supported by a ref on an ethological study. Are pigs bred in mixed or single gender pens in Sweden?

We have searched for, but not found reports from such studies. This was just speculation from our side thus we have now deleted this part. Castrates changed to barrows.

Line 131: please clarify the age at risk. Heavy production (Spain, Italy) are not at high risk.

Modified

Line 148-150: no need to report the directive table (and if you choose to do, please revise the intervals of square meters).

Revised

Line 151: this is a very interesting point. Please better explain how it is measured in Sweden (automated weighing of animals?) Also clarify how can the data in tab 1 be achieved: I assume that you put 10-13 pig into a pen after weaning and they remain until slaughtering (while in other countries they are moved and regrouped a few times). So, how can be that the more the weight increase the more the area per pig increase? It seems like the pen is progressively enlarged in some way… I disagree with the representation of EU trend: the space is progressively decreasing, given that small pigs are put into a large pen where they grow; once they are near the law limit they are moved into another larger pen, or the group is split.  

Answer: We have now clarified how farmers do to fulfil the Swedish requirement.

Line 199: revise the date

Revision made.

Line 204-212: explain how in Sweden the gas threshold are verified (have farmers some device for continuous gas monitoring? Do official veterinarians use some instrument for measurement?)

Asnwer: This is now explained in L 227-228.

Line 210: refer more clearly to Dir. 2008/120/EC

Revision made

Line 213: correct the date

Corrected accordingly

Line 220: clarify the connection between light and pig behaviour, avoid comparisons with hens and explain how in Sweden the compliance to 40 lux is verified

Text has been added in L 239-240

Line 255: consider EFSA guidelines

Answer: We are not sure what EFSA guidelines you are referring to. We have not been able to find any specific recommendations regarding length of feeding troughs in any report from EFSA. A conclusion from EFSA report on risks for tail biting is added in line 282-293.

Lines 274-280: such data on AMU are poorly informative, you should use dose-based methods (eg. DDDvet). Also the reference to EMA on veterinary sales is debatable (it includes sales for all livestock), you should use dose-based data on pigs specifically in other countries.

Answer: We agree on this. A new reference is added in Line 314-317 XX to address this issue where treatment incidence is based on DDDA:s. The EMA reference is removed.

A discussion on pig health in Sweden should also consider vaccination plans, level of biosecurity, and major pathogens occurring. Whether it is impossible to discuss all these aspect, they should at least be mentioned.

Answer: Information about pathogens, vaccination plans and biosecurity is now added in Line304-308. 

Line 316: I think in the meanwhile you published as conference proceedings.

References added.

Line 324-334: a current concern is the risk of pathogens in straw (eg. considering the risk for ASF, AD, etc.). Do you have in Sweden any procedure to assure a pathogen-free straw?

Reference added and line 380-389 has been added.

Line 335: there is a switch from point 3 to point 5

Changed accordingly

Line 355: recent publications rehabilitated the suitability of chains, although they need to be combined with other enrichment. Consider also the EU Working document suggesting a combination of suboptimal materials

The word “solely” added in Line 395 and he word “solely” added in Line 397 to a less clear-cut statement.

Line 374: growth promoters are banned in all EU states.

Answer: “antibiotics for preventative purposes” is now added in LN 421

Line 419: you have however a level of damage and the reader still may ask which is the treashold of acceptability. Moreover, it seems that most of the achievement is due to prevention, while scientific reports (eg. EFSA) also push on the importance of managing the problem (education of farmers, proper management procedures, number of sickbay areas per farm,…)

Hopefully, several modifications of the text have answered this concern.

Line 421: same unclear sentence than in the abstract.. are you suggesting that in other EU countries the legislation is applied like in Sweden or that the legislation is revised in order guarantee higher welfare standards (given that national Swedish legislation seems much more restrictive in some points)?

Now clarified in LN 392-396

Reviewer 5 Report

The paper gives an interesting comparison between rearing practices in Sweden, where intact tails are the norm, and other EU countries where tail docking is widespread. However, it is not always made clear which differences are believed to be most important and why (see below). In several places tail biting is stated unambiguously to be an indicator of poor housing conditions, although it is noted in places that other causal factors can exist. The first statement should therefore be qualified. Specific points: l13 and l25 and l43. Tail biting ‘can be’ an indicator of poor housing but can also be caused in good housing systems by other genetic, health, nutritional or management factors. L70. It is unclear what is meant by ventilation in this sentence. Does it mean that this is another relevant factor covered by Swedish legislation – if so, how? The use of ‘etc’ is not helpful. If there are other relevant factors in the legislation then describe them all. L114. Since the national prevalence is 3.2 (l89) does this suggest that the sample was biased? L129. So do you believe that castration is an important part of a non-docking system? L198. So is it a ban on fully slatted flooring per se, or the provision of substrate which is considered important? L212. So are there individual air quality aspects to be blamed for tail biting, or just air quality in general? L218. Since in winter natural daylength may be

Author Response

Reviewer 5

Comments and Suggestions for Authors

The paper gives an interesting comparison between rearing practices in Sweden, where intact tails are the norm, and other EU countries where tail docking is widespread. However, it is not always made clear which differences are believed to be most important and why (see below). In several places tail biting is stated unambiguously to be an indicator of poor housing conditions, although it is noted in places that other causal factors can exist. The first statement should therefore be qualified.

Clarifications in revision has been made.

Specific points: l13 and l25 and l43. Tail biting ‘can be’ an indicator of poor housing but can also be caused in good housing systems by other genetic, health, nutritional or management factors.

Text has been modified.

L70. It is unclear what is meant by ventilation in this sentence. Does it mean that this is another relevant factor covered by Swedish legislation – if so, how? The use of ‘etc’ is not helpful. If there are other relevant factors in the legislation then describe them all.

Answer: Now clarified in LN 70-73.

L114. Since the national prevalence is 3.2 (l89) does this suggest that the sample was biased?

Answer: No, as this is on farm level and the national average is on pig level. Once there is a tail biting outbreak a large proportion of the pigs are tail bitten. It may however be that farmers in this survey only report severe outbreaks and didn’t report when only minor damage occurred, damage that could have been found at slaughter.

L129. So do you believe that castration is an important part of a non-docking system?

Answer: Castration was not a main point here, thus this part has now been deleted.

L198. So is it a ban on fully slatted flooring per se, or the provision of substrate which is considered important?

Both, Pen function is more in line with the needs of the pig with partly slatted floor (i.e. place to go to the toilet, one to sleep and one to eat) AND provision of straw. This is now clarified in LN 211-214.

L212. So are there individual air quality aspects to be blamed for tail biting, or just air quality in general?

We have modified the text. Insufficient ventilation usually results in high levels of several gases. We do not speculate further in this topic.

L218. Since in winter natural daylength may be 

The text from the reviewer was truncated here, and it is not clear what should be changed.

Round 2

Reviewer 4 Report

The authors have revised the unclear parts. Last remarks:

line 422: is the biter maintained in isolation, sent to slaughter, euthanized or some attempts to reintroduce in the pen is made?
comment line 255 (see authors' response): see Doc XXIV/B3/ScVC/0005/1997

Author Response

”line 422: is the biter maintained in isolation, sent to slaughter, euthanized or some attempts to reintroduce in the pen is made?”

Text has been clarified, without going into too much details. The farmer may in a particular situation choose between several options, however here we describe the most common situation, and we have added the information that all pig compartments should have a separate hospital pen, according to the Swedish regulations.

“comment line 255 (see authors' response): see Doc XXIV/B3/ScVC/0005/1997”

The reference has been added as required by reviewer.

Thank you very much for all good comments.

Stefan Gunnarsson and co-authors.
